# Improved MQTT Secure Transmission Flags in Smart Homes

**DOI:** 10.3390/s22062174

**Published:** 2022-03-10

**Authors:** Asmaa Munshi

**Affiliations:** College of Computer Science and Engineering, University of Jeddah, Jeddah 21959, Saudi Arabia; ammunshi@uj.edu.sa

**Keywords:** smart home, secure MQTT protocol, Internet of Things, bi-directional transmission

## Abstract

In the current era of smart homes and smart grids, complex technical systems that allow for the automation of domestic functions are rapidly growing and becoming more widely available. A wide range of technologies and software applications are now available for use in smart homes, and many of them are free to use. They allow for communication between home appliances and their users, as well as the automation, monitoring, and remote-control capabilities of home appliances themselves. Unfortunately, a lot of previous research ignored security issues involving the great attention to detail of the data in a transmission session within the devices in smart home architectures, which is why this study proposed smart grid secured transmission flags suitable for preventing every bit of data transmission in a smart home. Secure Message Queueing Transport Protocol (MQTT) in Internet of Things (IoT) Smart Homes protocols was utilized; an experimental testbed was designed with a prototype involving the process of a smart home system and the sequences of the data transmission. The evaluation of the proposed strategies has shown an improved bi-directional secure resource constraint strategy for the smart home within data packet transmission at 70 to 80 mbps over secure MQTT. A number of concerns, including technological barriers, difficulties, challenges, and future trends, as well as the role of users, have been presented in this study, among others.

## 1. Introduction

Smart homes deliver on the need to drastically improve families’ lives through socially appropriate and timely assistance to perceive, anticipate, and respond to actions in the home [1]. Furthermore, smart homes must communicate with smart facilities, smart devices, and their underlying protocols [2]. Surprisingly, the development of a smart home system necessitates the development of capabilities that can quickly cross an unknown line, ensuring that the system functions adequately and efficiently. An important protocol associated with the interaction of intelligent devices and facilities in a smart home is MQTT [3]. This is part of the technology that enables user and device-to-device interaction in the smart home.

The fact that the MQTT protocol is one of the standardized protocols used in the Internet of Things for communication within an end-to-end network context in which security considerations are crucial is the motivating factor for this research [4]. Furthermore, it was clear that many IoT developers used MQTT because of its features, such as its ability to run on a small amount of memory and less computing power, and its low bandwidth needs. Unfortunately, security and the integrity of data are still the main reasons why the protocol is not used in many networks. People can also make new versions of this protocol because they can download and use the source code. This makes it easy to make new versions from MQTT version 5 [5]. The MQTT protocol set up a subscriber–publisher interaction with the help of a broker in a transmission over TCP [6].

In light of this, the current research proposes the following hypothesis: “Transmission Flags” in a transmission session over the MQTT protocol are associated with responses to security events at varying degrees of sensitivity to detecting security issues. This can be justified by the fact that the MQTT protocol, which is used by the broker to facilitate data sharing between publishers and subscribers, allows data to be exchanged between them through the use of MQTT control packets. Essentially, publishers are the devices that generate data and publish it to the broker; the data is organized in the form of topics, which allows subscribers who have subscribed to receive notifications depending on the topics they have subscribed to. When communication between a subscriber and a publisher of a specific topic is to be established, the MQTT broker will act as an agent to receive the subscriber’s request and transmit it to the publisher, and vice versa [7]. On the MQTT protocol, there are a number of security concerns that arise throughout the transmission session [8]. Due to the fact that the MQTT communication protocol does not provide an appropriate data protection mechanism during transmission, it poses a threat to the privacy and integrity of data. The MQTT protocol also allows a third party to get into a transmission session and subscribe to all MQTT messages that are being sent around the time of the transmission [9].

Despite prior research work on the analysis of weaknesses and vulnerabilities in MQTT security [10], MQTT security solutions following one of the largest cyber-attacks [11], and the proposed security architecture for MQTT [12], there are still a number of security challenges that require further investigation. The justification for that lies with the challenging issues of a secure communication solution for the IoT since there were so many variables to consider. Some of these difficulties were caused by the vastness of classification and the number of devices that can be used in the IoT, as well as the limited resources available for dealing with security approaches. In general, the MQTT broker architecture operates as a middleman between a publisher and a subscriber in order to establish a link between the two. In order to maintain secure contact between subscribers and publishers, it is necessary to develop a good security technique that provides secure communication between both parties, which is the goal of this article.

## 2. Related Work

There has been a great deal of prior research work on smart homes, and it has been demonstrated that smart homes are equivalent to home automation to some extent, in that smart devices interact with one another through a communications protocol such as MQTT in order to share data. The term “Smart Home” could also refer to a home that has been enhanced by the intelligence of its devices rather than the intelligence of the house itself. Previous research work on the data transmission within smart homes has yielded some interesting results. Crucial to this is the work that has been critical in this regard, such as Gawanmeh et al. [13], who conducted research and analysis on the protocols that enable device communication in a smart home environment with the goal of determining which protocols are the most important in terms of performance hierarchy level. While using fuzziness to analyze the performance of protocols used in monitoring medical signals in the human body in smart homes, it was discovered that the heartbeat–respiration medium access control (HR-MAC) protocol and the IEEE 80.15.4 protocol were the most appealing features of smart home environments used for medical signal monitoring in the human body, despite the fact that they were the most expensive to develop and implement. This is even though the speed of transmission can be affected by a lot of different things. In a coordinated data packet transmission of a device-to-device network environment, the limit of dataflow through which the devices communicate with each other at 434 MHz and a baud rate of 19,200 bps was found to be 25 ms in a range of distances between the transmitter and receiver from 0.5 to 970 m [14].

Despite the fact that these studies demonstrate the importance of communication protocols, which are intended to ensure that devices are connected and share data in an appropriate manner, they do not assess the security of the data while it is being transmitted. The issue of security, on the other hand, was not raised. According to this, the vast majority of studies do not consider both communication performance and handling security at the same time. 

However, Sakthivel and Vidhya [15] reveal that by using a system of remote key management in the cloud and for aggregating encrypted data, the research has proposed a technique for providing security in the transmission of data between devices as well as decreasing the computational time required to produce an efficient encryption system. Although a certain level of acceptable security has been achieved, the computational efficiency has been reduced. MQTT protocol was used in a quest to ensure good performance in device-to-device communication because of its ability to operate with minimal computational resources. However, security challenges continue to be a concern with this protocol. That is why Bhawiyuga et al. [16] proposed a token-based authentication for MQTT, where a token is generated for each transmission session and sent to the user if they are valid. The user then sends their token during the connection establishment phase to the broker, who verifies its validity before allowing them to publish and subscribe to the required topics. If they are valid, the broker will grant the user access to publish and subscribe. In terms of computational operation, the authentication process is costly, this is a similar approach to the authentication method established on elliptical curve cryptography proposed by Yeh et al. [17] for remote users, where it was revealed that the cost of computation was high compared to other protocols. However, using an improved approach proposed by Rahman et al. [18], a key policy attribute-based encryption by Elliptic Curve Cryptography, yielded a modified MQTT protocol capable of delivering secure communication between end devices. This is a more lightweight multi-tier approach to secure communication between low-end IoT Nodes.

Furthermore, a lightweight authentication system for light-edge IoT devices has been proposed by Shahidinejad et al. [19]. The analysis of the scheme revealed that it shows superiority in attack resistance, communication costs, and time costs of the proposed protocol over existing techniques. A reliable and lightweight mutual authentication scheme that is based on the cumulative keyed hash chain for IoT smart houses has been proposed by Alshahrani and Traore [20]. The scheme is intended to authenticate the identity of the sender; the authors follow a cumulative key hash chain. The protocol uses automatic validation of internet security protocols and applications and Burrows–Abadi–Needham logic for validation. Andy et al. [21] highlighted various attack vectors and security analysis of the complete MQTT protocol, where it was revealed that searching for MQTT protocol on port 1883, the default MQTT broker port that does not use the TLS mechanism for security purposes, will find an available broker server as a loophole in MQTT communication where the attacker can try to attack.

It has been shown by Haripriya and Kulothungan [22] that MQTT is vulnerable to a spoofing attack in which an MQTT broker recognizes a malicious or spoof message packet because it cannot identify the difference between the normal message packets and the spoof messages. This is in line with the fact that when an attacker has access to MQTT client information, they can steal identities using a spoofing attack [22]. MQTT has been identified as being vulnerable to flooding attacks, in which an attacker repeatedly sends a high number of connection requests and responses, flooding the broker. This is a typical form of DoS attack against MQTT to take over all of the connections that the MQTT broker can manage by creating a high number of simultaneous connections to the server [21,22,23]. This means that an attacker can eavesdrop on all messages sent between clients by subscribing to restricted subjects. It was also revealed that MQTT has also been exposed to SYN flooding attacks when an attacker conducts a TCP-based attack in order to establish several half-opened TCP sessions that can deplete message broker resources [23]. As found by Andy [21], an attacker who catches and alters an MQTT packet can use this to establish a man-in-the-middle connection with an MQTT broker without passing through the attacker’s computer.

Considering the security challenges outlined by the previous research study on MQTT transmission session, this study proposed an enhanced adaptive mechanism for secure data transmission in a smart home. The justification for this lies with the research gaps outlined that are related to the MQTT vulnerability to malicious firmware, by identifying an endpoint requesting a firmware update using MQTT. In a similar vein, malware can be introduced into the system as a result of this. The ability to evade authentication mechanisms such as access control lists is another vulnerability in MQTT [24]. An attacker can deduce the users’ credential by looking at the packet during the traffic analysis phase of the authentication procedure. As with information leaks, an attacker got into the network and did traffic analysis to get clear-text data from the MQTT protocol’s transit information [24].

## 3. Research Methodology

The research methodology used in this present study includes the processes taken from conceptualizing a smart home to analyzing the smart home’s network system and evaluating the influence of MQTT on the system. The following are the components of the research methodology used in this current study:Formulating the conceptual smart home MQTT model is step one.Developing an algorithm for detecting secure MQTT flags in MQTT messages.Experimental Analysis, which includes the following steps:
Establishing a testbed for Publish/Subscribe Data with a MQTT broker in order to collect data. The primary requirement for this system is to provide the appropriate level of security. In order to publish data, the device/sensor must first subscribe to MQTT broker topics, then the broker must verify the device/identity sensors from the SSL certificate database before accepting the subscription. If the device/identity sensors have not been verified, the subscription will be rejected. After that, this study will record the transmission session for later analysis.Examining the data from the transmission session


### 3.1. Architectural Framework

Home automation systems must be able to sense, forecast, and respond to home events if they are to deliver on their promise of drastically improving the quality of life for families by responding appropriately and immediately to user demand [25]. That is why they are described as smart homes. This study considers MQTT as a smart home communication protocol (see Figure 1). 

A smart home was created from a non-smart home (see Figure 2). An unintelligent house might be regarded as a house that has to be reconfigured before becoming intelligent. Figure 2 depicts a large number of interconnected sensors that communicate with one another over the MQTT protocol in order to help visualize this.

The architecture is based on a broker, which serves as the primary system, and which contains servers, communication networks, workstations, and internal linkage to the marketing system. Thus, a broker serves as a gateway, receiver, and server, and thus is required in some situations. It is possible for clients to submit very short one-hop messages to the broker and also receive messages if they have subscribed to a certain subject through the use of the MQTT server. The MQTT broker will sit in the middle and then allow the client to communicate. Users can engage with the communication because it is contained within an application service. The interactive terminal makes it simple for people to operate and establish connection and engagement with smart gadgets in their homes, saving them time and effort. The client that is subscribing to the subject is connected to the MQ broker via an open TCP connection, MQTT.

This scenario describes an MQTT broker as a gateway, which can be described as a receiver or a server, that allows clients to submit very brief one-hop messages to the broker and also receive messages if they have subscribed to a certain topic, as described above. Essentially, the MQTT broker is positioned between the clients, which is a fundamentally sound arrangement (A, B, C, and D in Figure 1). The clients should be subscribing to the same events, and the MQTT broker should be assisting in the transmission of the tasks connected with the events through the use of the MQTT broker. What really happens is that these specific clients B, C, and D subscribe to the tasks that client A has already published to the broker, and in this instance, the transmission is done via TCP/IP, so all four clients will open a TCP connection to the MQTT broker to which they have subscribed to the task, and they will then receive updates about the topic that has been published. This is often done in the same way that the post office model is done. To put it another way, in this scenario, the broker acts as a sort of post office, with clients who can either send or receive letters and messages with the addresses to which the communications are to be sent. If a client sends a message, any one of the other clients may be able to receive the message depending on the tasks that are associated with the message.

The MQTT protocol is lacking in terms of security, which is a major drawback. Despite the fact that MQTT supports user authentication through the use of a username and password authentication scheme, clients can still log in using their own authentication scheme. However, one of the issues researchers have been grappling with over the last couple of years, particularly in the context of the Internet of Things, is the lack of standards in communications, particularly when it comes to security.

Data is exchanged between publishers and subscribers via MQTT control packets sent by the broker. Subscribers can subscribe to get notifications from publishers, which are devices that generate data and submit it to the broker. These data are categorized into subjects by the publishers [23]. MQTT was created for IoT-based machine-to-machine communication. This stands for ensuring integrity, availability, and confidentiality are standard security objectives for every system, and the IoT Smart Home is no exception. Data alteration, traffic analysis, and home monitoring and control are all examples of associated attacks that can be exploited. The two main characteristics are the system’s goals of setting out to improve industrial device maintenance and operations by optimizing and forecasting their proactive behavior.

Managing transmission risks is a function of ensuring that clients communicate with the internet when providing data to the system administrator’s backend. Client authentication and transport layer security are handled by a controller for hardware security supplied by this system, which proposes to communicate data via MQTT and to provide a controller in order to protect the hardware. This allows for a preliminary check, which is why this study conducted a preliminary study; the inclusion of security components in this smart home has no negative impact on the performance of the hardware [26].

Based on the proposed concept, the functional specification for a smart home MQTT model is presented in Figure 3. The primary requirement for a smart home system is to provide the appropriate level of security for the identification of the user who will log into the Smart Home System in order to view the status of embedded devices, as well as the capability to manage the machine-to-machine identification that connects to the MQTT broker for subscribing and/or publishing messages. Putting information out there requires an embedded device communicates with a MQTT broker, and the data exchanged between the two is encrypted. The interactions occur between an MQTT broker and a smart home system.

The smart home scenario provided in this study includes a creation element, into which the user will enter their username and password in order to log into the system, as shown in Figure 1.

That is the event that sets off the chain of events. If the user has successfully authenticated themself, the login page will redirect them to the system dashboard page. This is followed by verifications, in which the system will confirm the user’s identity in the database by comparing the username with the associated credential (if applicable). Each smart home device will be represented by a statue on the system. The dashboard page will give a summary of the current status of all smart home devices obtained from the MQTT broker. The user has the ability to add, reset, and adjust all smart home devices that are connected to the system. The Event System will indicate the status of each smart home device and save any administrative actions taken on it. The dashboard page will give a summary of the current status of all smart home devices obtained from the MQTT broker. One can add, remove, reset, and adjust Smart Home System devices and sensors with success.

With the MQTT broker, data communication between smart home devices and sensors is enabled through the use of publish/subscribe data of the sensor/device in question. If you subscribe to one of the MQTT broker topics, the broker will verify the device/sensor identification from the SSL certificate database before accepting the subscription and publishing the data. If the device/sensor identity has not been validated, the subscription will be disallowed. The device verification process checks the authenticity of the device/sensor by comparing the device/sensor name with SSL certificates in order to authenticate its identity for the purposes of publishing and subscribing.

It is anticipated that users will connect into the smart home application using their username and password, either on their mobile device or through a web-based platform, and that there will be two database-authentication situations, namely: If the user is authenticated, he or she will be able to access the system and log in successfully. If the user is not authorized, he or she will be unable to access the system; the login attempt will be rejected, and an error message will be displayed. After logging into the system, the dashboard will display the current state of all devices and sensors, as well as an icon for each device and sensor’s settings and an icon for adding a new device or sensor. In addition, the add icon allows the user to connect a new home device/sensor to the server by specifying the device’s ID, its name, and its location, and when the server gets a new device, it creates a certificate for it using the MQTT SSL certificate. The database settings icon allows the user to make changes to, remove, or reset a device or sensor that is currently linked to the system. When a user removes a device or sensor from a system, the SSL certificate associated with that device or sensor is removed from the MQTT broker database. A confirmation notice will be shown to the user after each step (Add, Setting, and Remove) to ensure that the changes have been saved. The devices/sensors scenario represents a freshly inserted device in the MQTT broker database that has been given a name and is attached to an SSL certificate. When a device or sensor sends a subscription or publish request to a broker, the broker will validate the device or sensor by authenticating the device or sensor’s SSL certificate from the database; if the device or sensor is authenticated, access is granted; if it is not authenticated, access is denied. When a device or sensor is deleted, the SSL certificate associated with that device or sensor is also removed from the SSL certificate database on the MQTT broker server (see Figure 4).

### 3.2. Algorithm Development

Developing a conceptual architecture for the procedures necessary to tackle the MQTT security problem is the central point of the proposed algorithm development strategy. The suggested improved MQTT secure transmission flags for usage in smart homes are based on two TCP Header Flags: TCP Explicit Congestion Notification (ECN-Echo) and Congestion Window Reduced (CWR) associated duties. That is, it was designed to be utilized in a similar manner to how TCP utilizes various flags to control flow. This is because TCP does not provide security flags, while MQTT utilizes TCP transmission sessions. As a result, this study takes the approach of including security flags in MQTT transmission. As seen in Figure 5, the proposed notion is sandwiched within TCP flags from which an encryption flag for data in a MQTT transmission session would be generated. Additionally, this flag will be associated with the embedded Transport Layer Security (TLS) protocol, Secure Sockets Layer (SSL), a security mechanism for authenticating users and preventing unauthorized user access to MQTT.

In the proposed algorithm (see Algorithm 1), the first step is to initialize transmission flags, where an authentication based on topic, subscriber, and publisher were defined from lines 2 to 4, which ensures that the connections and interaction are established based on topic, subscriber, and publisher. From lines 5 to 7, everything related to monitoring the security is attached to a defined flag. From lines 8 to 9, this is where the request for the security flag was cranked up to the top. Later, the connection between individual nodes of sensors from line 16 is formed with brokers, and the server receives a new device and creates a new certificate for it in the MQTT SSL certificate database, which is shown in lines 10 to 14. The operation is carried out by which the MQTT broker confirms that the message will appear and has applied TLS and SSL on MQTT based on inspecting the MQTT packet matrix with all of the options available. When using a client-to-broker technique, the payload of the message is encrypted during the communication between the client and the broker, ensuring that the message is secure. The broker has the ability to decrypt the message before distributing it, resulting in all subscribers receiving the unencrypted version of the message. For situations where TLS and SSL are working together with the MQTT flags, they will provide a defense against sensitive data.
**Algorithm 1** Secure MQTT Flags **Initialize** *Transmission Flags, packet ID(pid)*1:**Begin**2:
 **if** topic(*t*) name and width ← ∈ {packet_set}‘ **then**3:

  *pid* = trans_ps(*pid*)4:

  request (subtribe, publish) as key/value pair5:

  
**then**
6:createSecureSubtribePublish (*SSP*);7:   Input: *retainFlag*, *QoS*, *dubFlag*, A string description, unsigned int value, One address recipient address8:     R ← Is the list of requests *SSP*9:     NR ← The new request object *SSP*10:   secureFlag(*SSP*Name, *SSP*List)11:    Set NR.complete ← true12:       **for** each NR.complete in SSPList **do**13:        n,r = Encryp.index of NR.complete14:         set EncryptNR.complete at x,y 15:     **end for**
16:  Secure_MQTT = (secureFlag ← ∈ *SSP*17: Apply TLS&SSL = *SSP*(*x,y*) + TLS(*x,y*) + SSL(*x,y*)18:Examine MQTT packet matrix with Xor matrix received19:**return** secure selected MQTT relevant feature20:**end**

## 4. Experimental Analysis

The experimental testbed was constructed using the “OTA WeMos D1 CH340 Wi-Fi Development Board ESP8266,” the “Elegoo EL-KIT-001 UNO R3 Project Complete Starter Kit,” and a Raspberry Pi B+, all of which were purchased from Amazon. When it comes to technology, the “WeMos D1” is a “ESP8266 Wi-Fi-based board” that operates on a “Arduino layout” and has an operating voltage of 3.3 V. This is one of the most important tools used in the creation of smart homes, since it is required to set up a board of construction for development. Additionally, the “2-Elegoo EL-KIT-001 UNO R3 Project Complete Starter Kit” is available on Arduino kits, which contain an “Arduino Uno board” as well as additional components that are required to imitate electrical programming. With these kits, the project will be able to construct smart homes with the Raspberry Pi B+ serving as the backbone. The Raspberry Pi is a small, low-cost computer the size of a credit card that connects to a computer monitor or television and can be controlled with a regular keyboard and mouse. It is capable of doing all of the functions normally associated with a desktop computer. In order to communicate with other devices, the Raspberry Pi must be connected to the internet.

A mobile interactive interface with the smart house MQTT model was created after the setting had been constructed (Figure 6). Icons and buttons, among other things, are used to allow the user to interact with a product or device while engaged with the system. UI refers to the frontend page that is displayed to the app user when the app is opened and allows the app user to interact with the backend of the app. A smart home app displays four screens for users during the design interaction; these screens are as follows: a login page for the user, a dashboard screen for the user to monitor system status, a screen on which the user can adjust, reset, or delete a device/sensor, and finally a screen on which the user can add a new device/sensor with its associated room name.

When it came to communication between devices and apps, the MQTT Protocol was used to operate over the transport standard for communication. Open SSL, which is a protocol for delivering encrypted connection between devices and applications, was also used to run the design. The implementations were carried out using Eclipse Paho, whereas Eclipse Mosquitto is a lightweight server implementation of the MQTT protocol that is suited for a wide range of circumstances, from full-power workstations to embedded and low-power devices. It was necessary to use Visuino to read and program Arduino boards because it was supported. Ubuntu Mate was selected as the final operating system because it is a stable, user-friendly operating system with a highly customizable desktop experience.

This project simulates the smart home system devices by configuring Arduino sensors and Raspberry pi to communicate with an MQTT broker to exchange the data of a smart home; this data will be a plain text data because of non-encryption methodologies in MQTT protocol. After that, the research uses SSL protocol to encrypt the data that is transmitted between the sensors and broker by applying encryption methodology to transmitted data. At first, it was a matter of connecting an Arduino board to a Visuino computer in order to configure the board set to communicate with a Wi-Fi network and a MQTT broker. With the Arduino IDE, you may configure the sensors’ settings by writing C++ code and then uploading it to the Arduino board, which is connected to the appropriate sensor. This is followed by the configuration of the sensor’s subscribing to and posting data to the relevant topic, which are both optional. Following that, the SSL protocol is applied to each sensor, and the SSL certificate is uploaded to the MQTT server together with the sensor in question.

## 5. Presentation of Experimental Analysis Results and Discussion

An experimental setup was created with secure MQTT flags enabled. This section presents and discusses the results of the experiment, which included the evaluation of the experiment. Simulations of the smart home prototype that was built and can be accessed through a mobile app were carried out by running several transmission sessions involving the transmission and capture of the transmission session data flow using the “network protocol analyzer” provided by TShark Software. For each transmission session, the sensor boards connect to the internet via Wi-Fi in order to publish and subscribe data to its related topics in an MQTT broker. The data sent was encrypted, and the MQTT broker received the data, decrypted it using the appropriate SSL certificate from the SSL database, and then published/subscribed to its related topics in the MQTT broker.

The connection was set out to deny broker data that can receive data from any board whose SSL Certificate is not verified in the SSL database. Additionally, publishing or subscribing to the broker topic will be denied. The simulation environment includes nodes that constantly transmit data. Because of the distributed nature of the nodes, the number of nodes that send data are random. The node sends the same request more frequently and at a higher rate. Finally, the data was captured and the flags of secure MQTT were compared with those of normal MQTT, and the results of the comparison show that secure MQTT outperforms regular MQTT in terms of performance. The duration of each time frame is fixed while the publish-subscribe messages change.

In order to determine the degree to which the data in transmission is secure in terms of unveiling and reviling the content of the transmission payload and header data, the proposed scheme captured those transmission sessions’ information and analyzed them. The false-positive ratio of unveiling and reviling the content of the transmission, and precision rate to verify and validate the performance unveiling and reviling the content of the transmission were calculated and presented.

The verification rate within various transmission sessions performed during the experiment are presented (see Figure 7). Many transmission scenarios were recorded in order to assess the importance of verifying the integrity of messages and to compare the effectiveness of different transmission methods. Within the four scenarios presented (see Figure 7A–D), they indicate the transmission within the connection for each session. Figure 7A shows secure MQTT transmission at a rate of 40 to 60 mbps, and the transmission session data associated with the header flags is presented in Table 1. The difference between Figure 7B–D is that in all cases, the details transmission flag is detected within a reasonable range of data rate. The publisher and subscribers send information to the broker via the internet. The design is such that both the publisher and the subscriber could be from untrusted MQTT clients, which could be potentially harmful. The checking is done in relation to the data integrity checks that allow the transmission to ensure that third parties did not modify any content of the MQTT message runs by subscribing and receiving a message from published command in port 8884, then the communication between server and client is secured and the data is encrypted. When comparing transmission over secure MQTT to transmission over conventional MQTT, an important result was obtained (see Figure 7; *y*-axis is the transmission rate in mbps).

Practically all of the transmission sessions had secure MQTT present obscured messages with a higher obscureness than with normal MQTT messages (See Figure 7). The transmitted data between the smart home devices and MQTT server are encrypted, hence the publish/subscribe to MQTT broker topics are verified. The inability of anonymous devices from publishing and subscribing on MQTT broker topics was subject to this verification. The entire transmission session data captured is presented in Table 1.

The summary of the transmission session captured results presented in Table 1 was carried out within some different ranges of devices where the subscribe and publish packet data within the broker received the amount of data ranges for each transmission session. The data packet data is received by the devices. The results demonstrate that the MQTT Publish packets from (V1 to V3) verify the contents of the packet, which is encrypted from the beginning of the transmission until the end. Results reveal that the MQTT Publish packets are verified by the contents of the packet, which is normally added to the payload at the beginning of the transmission, while the receiver of the packet can validate the integrity of the data by validating the contents of the packet. This validation ensures that the message has not been tampered with by a malicious third party after it has been transmitted. This has an influence on home automation systems since it ensures that every piece of data, whether sensed or forecasted, is protected. Additionally, the response to home events should be delivered on their promise in a secure manner, with each device responding correctly to the device that it is immediately adjacent to. MQTT, the smart home communication protocol, uses security flags to intelligently determine whether or not a secure channel is present. When all of the networked sensors communicate with one another using the MQTT protocol, the data is protected against interception.

Typically, the findings of the study are based on the broker, which serves as the core system and incorporates servers, communication networks, workstations, and an internal link to the marketing system, among other components. Because it is required in any eventuality that may arise during the transmission session, broker security is extremely important. The security of the client is also critical, because messages sent to the broker by the client in certain circumstances must be protected. That is, the message sent to subscribers should be secure to a certain extent. The security provided by the MQTT broker is also very important and critical to the client’s ability to communicate. Due to the fact that it is housed within an application service, transmission within any link is critical.

MQTT brokers, which can also be represented as either a receiver or a server, allow clients to send extremely brief one-hop messages to a MQTT broker in this situation. Messages can be received by clients if they have subscribed to a specific topic, which is referred to as either a receiver or a server in this scenario, respectively. When it comes to security, the MQTT broker’s position between clients is critical, and it is this position that must be taken into consideration. In order for the tasks associated with a subject to be successfully transmitted, both the clients subscribing to the subject and the MQTT broker must establish a secure transmission session. A secure transmission session is required by the MQTT broker to which the subject has been subscribed, and the broker will only receive updates about the topic that has been published in a secure transmission session as a result of this discovery. In order for the MQTT protocol to function properly, it must be connected to a secure communication session. The broker facilitates data exchange between publishers and subscribers by sending MQTT control packets to both of their addresses. The ability to subscribe to receive notifications from publishers, which are devices that generate data and submit it to the broker through a secure transmission session, is available to subscribers. In the case of publishers, these are devices that generate data and transmit it to the broker through a secure transmission session.

A functional specification for the MQTT model for the smart house has been developed on the basis of the notion of secure transmission session that has been proposed. The most important requirement for a secure transmission session is the provision of an acceptable level of security for user identification during the transmission session. Examples of secure transmission session requirements are provided in the following section. In order to enter information into the system, an encrypted transmission session must be established. In order for the MQTT broker to be able to use publish/subscribe data, it is necessary for it to employ secure transmission session verification. If the verification procedure examines the authenticity of the transmission session in order to confirm its identity for the purposes of posting and subscribing to content, then it is necessary to use a secure transmission session to protect the information transmitted.

The findings of this study have important ramifications since they demonstrate that authentication based on subject, subscriber, and publisher defined for MQTT is unquestionably set up to ensure that connections and interactions are made solely on the basis of subject, subscriber, and publisher. It is because of this that it is safe. It was because of this that the conclusions of this study were achievable. A consequence of the study, it was determined, was that for each transmission session, the study setup concluded that a transmission was independent of any other transmission despite the fact that each transmission session used a different data range and the same procedure as the previous one.

The study’s additional implication is that the concept of secure transmission sessions can be applied to a functional definition for the MQTT model for the smart home, which would be useful for future research. It has been determined that the most important criterion for a secure transfer is an acceptable level of security for user identification during the transmission session. Before any data can be transmitted, it is necessary to establish a secure communication connection with the intended recipients. A secure transmission session verification is required by the MQTT broker in order for them to be able to publish and subscribe to data in their transmissions to be successful. To put it another way, in order to post and subscribe to content, a secure transmission session must be utilized to ensure that the transmission is legitimate before it can be used.

Finally, this research has been able to demonstrate that clients can utilize MQTT brokers in a secure manner in order to transfer data to another MQTT broker, and it has also been able to assess the quality of the security in the data transmission. The location of MQTT brokers in respect to their clients must be taken into consideration in order for safe transmission to be established between them. All transmission notes in a transmission session that are involved in sending tasks that are related to a subject must be encrypted in order to produce a transmission session that can be monitored and confirmed. This is one of the most significant research accomplishments of this study.

Although these implications have been emphasized, the study has several limitations, particularly in terms of the extent of the network within the smart home, which must be addressed. Typically, this is done to ensure that all of the tasks linked with a subject are effectively completed. A MQTT broker that has subscribed to the subject will require a secure transmission session in order to get updates about the topic and will only receive updates about the topics that have been published in a secure transmission session. As a result, future studies should take into account the vast network environment.

## 6. Conclusions

This study has recognized that in the contemporary era of smart homes and intelligent, complex, and technical solutions that automate domestic operations, there is a need for ensuring that the security of the system is well addressed. Considering that there is ever expanding use of the IoT, the study conceptualized a smart home that can take advantage of a wide range of technologies that are currently used. That is why the analysis of data within devices in an intelligent environment is crucial. An experimental setup of a smart home is needed where users can communicate with their computing household appliances that are automated in order to monitor and manage them efficiently. Hence, the study provided a smart environment with the provision of secured transmission flags ideal for securing every bit of data transmission within devices in smart home architectures, MQTT was employed, and an experimental testbed with a prototype smart home system and data transmission sequences was built and performed. The evaluation of the proposed techniques indicated a better interaction and highly secured smart home resource that shows MQTT can protect packets that are checked by packet content, which increased security.

An important communication variable is “Transmission Flags inside a transmission session of sensors in a Smart Home”. This shows that the influence of everything that occurs while sensors are sharing data in a network conversation could be determined by transmission flags. For each communication transmission session under consideration, this study set up a transmission session testbed and recorded the events that occurred inside various ranges of devices. The data ranges for each transmission session were recorded. This present study’s results only cover data packets that can be received by devices via MQTT and secure MQTT and have a variable payload length; however, future studies can analyze data packets that can be received by devices via MQTT and secure MQTT with an unlimited payload length.

## Figures and Tables

**Figure 1 sensors-22-02174-f001:**
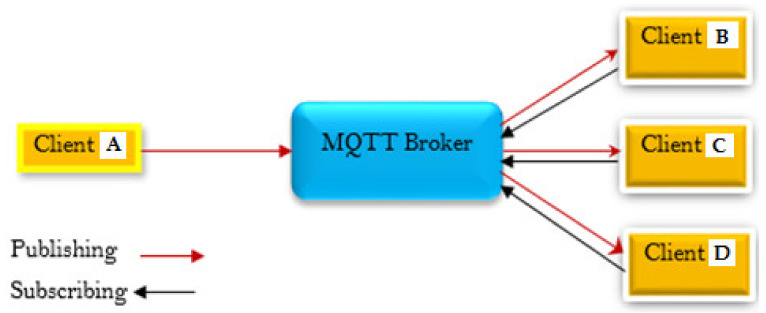
MQTT publish-subscribe model.

**Figure 2 sensors-22-02174-f002:**
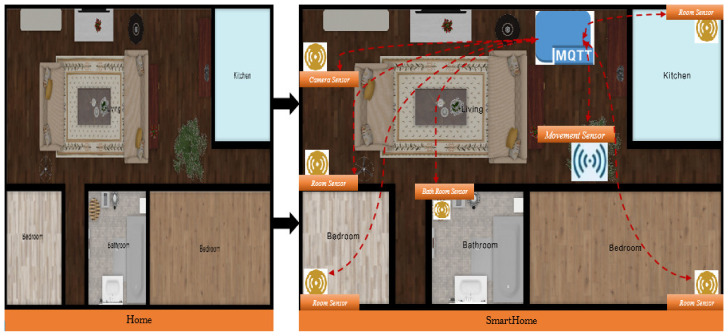
The conceptual smart home MQTT model.

**Figure 3 sensors-22-02174-f003:**
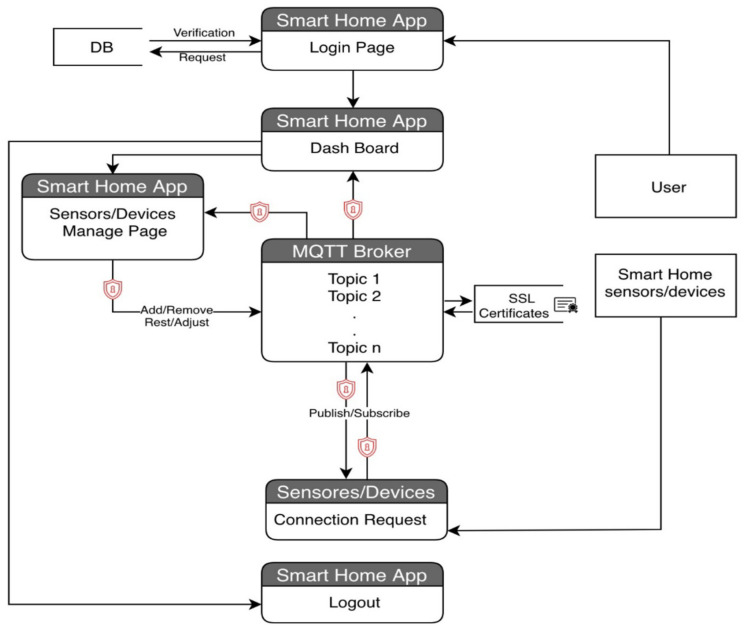
The functional requirement for smart home MQTT model.

**Figure 4 sensors-22-02174-f004:**
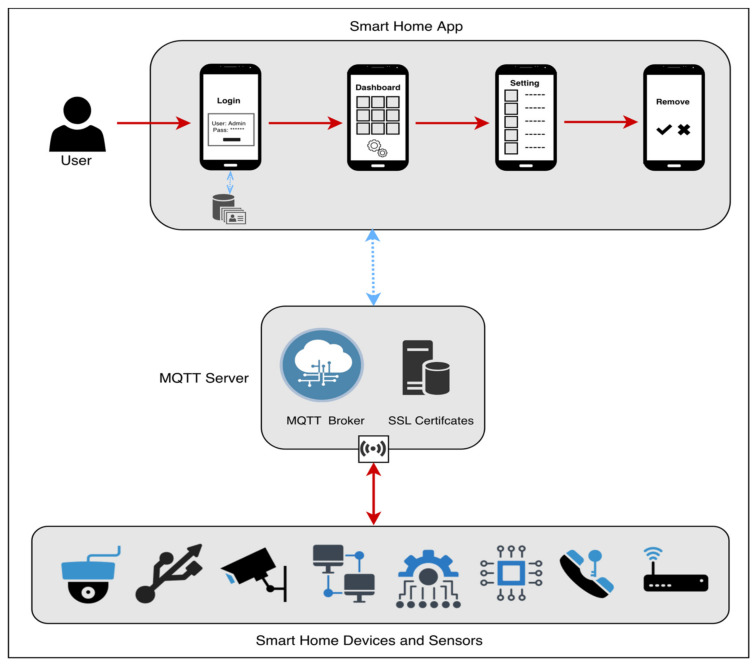
The conceptual smart home MQTT model.

**Figure 5 sensors-22-02174-f005:**
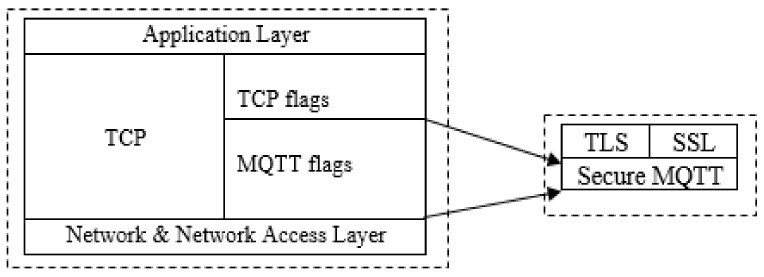
The secure MQTT model.

**Figure 6 sensors-22-02174-f006:**
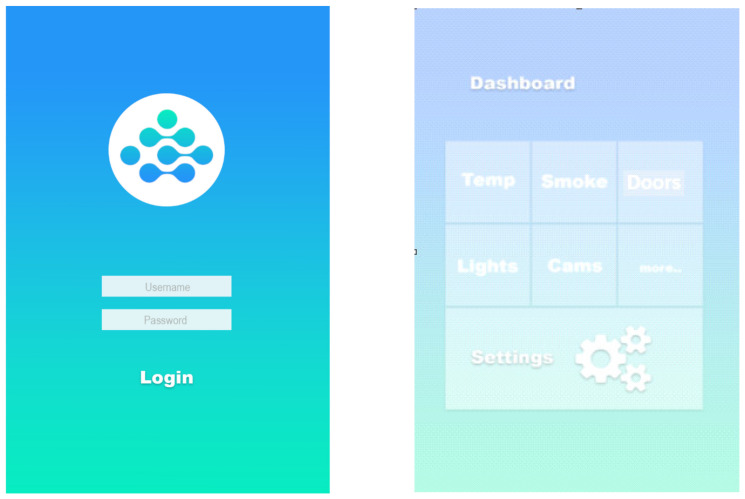
The interaction views with the smart home MQTT model.

**Figure 7 sensors-22-02174-f007:**
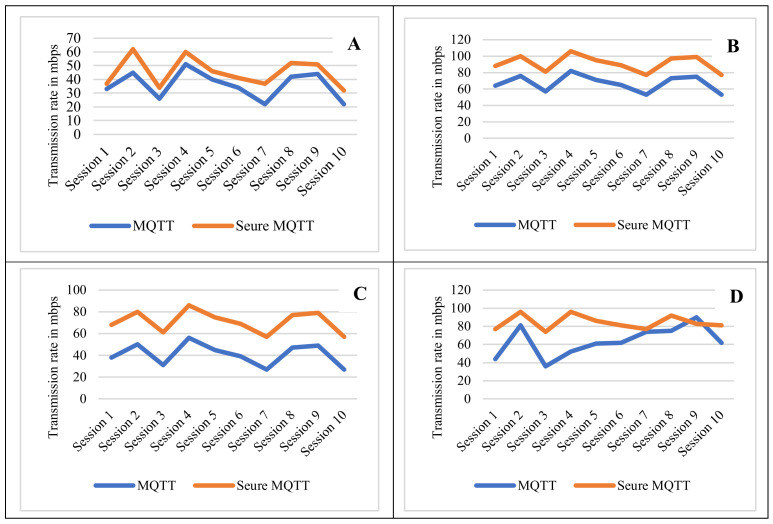
The verification rate of obscureness.

**Table 1 sensors-22-02174-t001:** Transmission sessions captured data.

Ses	t_s_/m	t_t_/kb	a_s_/kb	t_a_/s	t_d_/ms	tm/ms	a_d_/ms	f_t_/ms	a_p/_ms	V_1/_kb	V_2/_kb	V_3/_kb
S1	7.1	406.6	74.6	25.102	288.6	50.6	282.6	21.6	267.6	64.6	10.544	6.813
S2	7.1	406.6	266.6	27.202	302.6	56.6	85.6	26.6	65.6	64.6	21.68	7.3281
S3	7.1	406.6	406.6	29.176	65.6	55.6	62.6	31.6	37.6	64.6	29.8	7.628
S4	106.6	406.6	406.6	29.183	67.6	59.6	62.6	31.6	37.6	64.6	29.8	7.627
S5	976.6	406.6	108.6	14.07	1366.6	52.6	79.6	31.6	54.6	64.6	12.516	7.393
S6	666.6	406.6	406.6	29.187	68.6	58.6	62.6	31.6	37.6	64.6	29.8	7.627
S7	7.1	406.6	406.6	29.184	80.6	44.6	62.6	31.6	37.6	64.6	29.8	7.627
S8	666.6	406.6	406.6	29.183	74.6	54.6	62.6	31.6	37.6	64.6	29.8	7.627
S9	666.6	406.6	406.6	29.183	74.6	54.6	62.6	31.6	37.6	64.6	29.8	7.627
S10	666.6	406.6	406.6	29.183	74.6	54.6	62.6	31.6	37.6	64.6	29.8	7.627

Ses = transmission session, ts(m) = ranges, tt = initial transmitted data (kb) as = second transmission data (s) tx = transmission, td = third round, tm = fourth transmission time, fd = second to last transmission time, ap = last transmission time, V1 = secure MQTT data packet, V2 = secure MQTT data packet, V3 = secure MQTT data packet size in bytes.

## Data Availability

Not applicable.

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
