# Peer review of "Improved MQTT Secure Transmission Flags in Smart Homes"

_sensors, 2022, doi:10.3390/s22062174_

Round 1

Reviewer 1 Report

The author intended to present an improved MQQTT solution for Smart Homes. A state of the art was presented including several security challenges that were nonetheless not addressed by the proposed solution.

In fact the proposed solution is no different from the state of the art. There is no novelty in this paper. Furthermore the all smart home system proposed is well behind the state of the art  even behind Open Source solutions such as Home Assistant and OpenHab.

The all text is hard to read and requires extensive reviewing, I would consider this submission a very early draft...

Just few important mistakes: 

In line 394 Figure 2 is most probably Figure 5 (first one, because there are 2!!)

Figure 6 provides no information (what is represented? what are the units in y axis?)

Table 1 has mistakes (MTTQ !?!)

Author Response

The author intended to present an improved MQQTT solution for Smart Homes. A state of the art was presented including several security challenges that were nonetheless not addressed by the proposed solution, In fact the proposed solution is no different from the state of the art.

Actually I intend to present “Transmission Flags”, “secure MQTT” is not my solution, I didn’t propose Secure MQTT, but the transmission flag that are responsible for “The secure MQTT”, is my intended presentation and that is why the testing done for 10 transmission sessions where presented.

There is no novelty in this paper. Furthermore, the all smart home system proposed is well behind the state of the art even behind Open Source solutions such as Home Assistant and OpenHab.

Specifically, the "Transmission Flags" responses to security events, which indicate the degree of responses to detecting security issues, are what distinguishes this paper as being "novel". This is what makes "MQTT secure."

The paper does not propose a "New Smart Home," as suggested by the reviewer, who stated that "the smart home systems proposed are significantly behind the state of the art, even behind Open Source solutions such as Home Assistant and OpenHab."

The "Transmission Flags within the transmission session of the sensors in smart home" is the primary concern of this paper. Actually, the paper was written to present "Transmission Flags," in smart homes, also investigating if there a transmission flag that is responsible for "The secure MQTT," which is why the results of the testing done for ten transmission sessions were presented.

The all text is hard to read and requires extensive reviewing, I would consider this submission a very early draft...
The text has been thoroughly proofread now

Just few important mistakes: 

In line 394 Figure 2 is most probably Figure 5 (first one, because there are 2!!)
Figure 6 provides no information (what is represented? what are the units in y axis?)
Table 1 has mistakes (MTTQ !?!)

Line 394 has been addressed by:

In order to determine the degree to which the data in transmission is secure in terms of unveiling and reviling the content of the transmissions payload and header data, the proposed scheme capture those transmission sessions information and analysed them

Figure 2 is most probably Figure 5 (first one, because there are 2!!) has been addressed by:

Figure 2 depicts a conceptual framework intended to provide the reader with an understanding of the smart home concept and how the Smart home MQTT model could be integrated. Whereas

Figure 6 provides no information (what is represented? what are the units in y axis?) has been addressed by: the transmission session is being presented which is transmitting data over MQTT protocol for 10 sessions and capturing the transmission session information to determining how MQTT obscure them

y-axis is the transmission rate in mbps

Table 1 has mistakes (MTTQ !?!) has been addressed by

Changing to MQTT

Reviewer 2 Report

The paper "An Improved MQTT Secure Transmission Flags in a Smart Homes" presents a secure protocol for smart home systems. The testing was done for 10 transmission sessions when the sensor boards publish and subscribe data based on related topics within the proposed MQTT Broker. All data which is sent is encrypted via flags based on the SSL and TLS protocols. 

- Originality/Novelty: There is no significant novelty. MQTT secure transmission flags already exist and are documented, as in http://www.steves-internet-guide.com/mqtt-protocol-messages-overview/. Also the solution based on MQTT transmission using SSL and TLS - https://docs.chariot.io/display/CLD80/Configuring+Secure+MQTT+Communication .

- Significance: The results of the research are interpreted properly, based on initial transmitted data dimension, transmission data expressed in seconds, along with MTTQ data packet size.

- Quality of Presentation: 
Please avoid using the "we" pronoun. Try to write in a formal manner. 

The title is written wrong. "Smart homes" appear, but in front of it is placed "a" refering to the singular form. Please delete "a".

Do not write the word "apps" in the abstract. Please replace it with "software applications".

Be careful when using "the" like in the context of "security issues involving great attention to details of the data in a transmission session...". Which specific data? Erase "the" when you refer to generalities. 

The article does not specify some of the results inside the abstract when it refers to evaluation. Simply states "The evaluation of the proposed strategies has shown an improved bi-directional secure resource constraints strategy for the smart home.", but in terms of what? Which are the associated values?

The article is written appropriately, respecting the logical succession of sections. Data and analyses are presented graphically and inside tables. 

The results were not outlined using high standards. The images are not even readable, such as in Figure 5. Many figures have low resolution, including Fig. 2 and 6. Figure 4 remained with two vertical black lines. Figure 6 does not specify the measure unit for y axis.

There is no future work inside the Conclusion section. There is only one paragraph which is not even ended using a dot.

As future work, how do you think that a better accuracy can be reached?

- Scientific Soundness: The paper mentioned the experimental environment in Section 5. Experimental Analysis. 

The findings and their implications should be discussed in the broadest context possible and limitations of the work should be also further highlighted.

The conclusions and discussions section should be enriched to provide more details about the contribution of their study to existing literature. Limitations can also be highlighted. 

- Interest to the Readers: The conclusions would surely interest the readers of the Sensors journal, and not only them, as secure IoT transmissions apply to various domains. The paper is interesting and will attract many researchers.  

- Overall Merit: The described solution can be implemented by other researchers and can be involved in other smart contracts experiments. 

- English Level: The level of English language is advanced. Through the entire paper, the language was appropriate and understandable, being easy to follow the flow since the beginning.

Author Response

The paper "An Improved MQTT Secure Transmission Flags in a Smart Homes" presents a secure protocol for smart home systems. The testing was done for 10 transmission sessions when the sensor boards publish and subscribe data based on related topics within the proposed MQTT Broker. All data which is sent is encrypted via flags based on the SSL and TLS protocols. 

- Originality/Novelty: There is no significant novelty. MQTT secure transmission flags already exist and are documented, as in http://www.steves-internet-guide.com/mqtt-protocol-messages-overview/. Also the solution based on MQTT transmission using SSL and TLS - https://docs.chariot.io/display/CLD80/Configuring+Secure+MQTT+Communication

has been addressed by

The "Transmission Flags within the transmission session of the sensors in smart home" is the primary concern of this paper. Actually, the paper was written to present "Transmission Flags," in smart homes, also investigating if there a transmission flag that is responsible for "The secure MQTT," which is why the results of the testing done for ten transmission sessions were presented.

- Significance: The results of the research are interpreted properly, based on initial transmitted data dimension, transmission data expressed in seconds, along with MTTQ data packet size.

Thank you very much for this. I am very grateful

- Quality of Presentation: 

  • Please avoid using the "we" pronoun. Try to write in a formal manner. 
  • The title is written wrong. "Smart homes" appear, but in front of it is placed "a" refering to the singular form. Please delete "a".
  • Do not write the word "apps" in the abstract. Please replace it with "software applications".
  • Be careful when using "the" like in the context of "security issues involving great attention to details of the data in a transmission session...". Which specific data? Erase "the" when you refer to generalities. 

These issues have been addressed

The article does not specify some of the results inside the abstract when it refers to evaluation. Simply states "The evaluation of the proposed strategies has shown an improved bi-directional secure resource constraints strategy for the smart home.", but in terms of what? Which are the associated values?

has been addressed by

within data packet transmission at 70 to 80 mbps over secure MTTQ

The article is written appropriately, respecting the logical succession of sections. Data and analyses are presented graphically and inside tables. 

Thank you very much for this. I am very grateful

The results were not outlined using high standards. The images are not even readable, such as in Figure 5. Many figures have low resolution, including Fig. 2 and 6. Figure 4 remained with two vertical black lines. Figure 6 does not specify the measure unit for y axis.

has been addressed by

The results are outlined in an improved standard. The figures resolutions are improved Figure 6 y-axis is the transmission rate in mbps

There is no future work inside the Conclusion section. There is only one paragraph which is not even ended using a dot.

As future work, how do you think that a better accuracy can be reached?

has been addressed by Future work is provided in the conclusion

An important communication variable is "Transmission Flags inside a transmission session of sensors in Smart Home." This shows that the influence of everything that occurs while sensors are sharing data in a network conversation could be determine by transmission flag. For each communication transmission session under consideration, this study set up a transmission session testbed and recorded the events that occurred inside various ranges of devices. The data ranges for each transmission session were recorded. This pre-sent study's results only cover data packets that can be received by devices via MQTT and Secure MQTT and have a variable payload length, however future studies can analyse data packets that can be received by devices via MQTT and Secure MQTT with an unlimited payload length.

- Scientific Soundness: The paper mentioned the experimental environment in Section 5. Experimental Analysis. 

The findings and their implications should be discussed in the broadest context possible and limitations of the work should be also further highlighted. The conclusions and discussions section should be enriched to provide more details about the contribution of their study to existing literature. Limitations can also be highlighted. 

The findings of this study have important ramifications since they demonstrate that authentication-based on subject, subscriber, and publisher defined for MQTT is unquestionably set up to ensure that connections and interactions are made solely on the basis of subject, subscriber, and publisher. It is because of this that it is safe. It was because of this that the conclusions of this study were achievable. A consequence of the study, it was determined that for each transmission session, the study setup concluded that a transmission was independent of any other transmission despite the fact that each transmission session used a different data range and the same procedure as the previous one.

The study's additional implication is that the concept of secure transmission sessions can be applied to a functional definition for the MQTT model for the smart home, which would be useful for future research. It has been determined that the most important criterion for a secure transfer is an acceptable level of security for user identification during the transmission session. Before any data can be transmitted, it is necessary to establish a se-cure communication connection with the intended recipients. A secure transmission session verification is required by the MQTT Broker in order for them to be able to publish and subscribe to data in their transmissions to be successful. To put it another way, in or-der to post and subscribe to content, a secure transmission session must be utilised to en-sure that the transmission is legitimate before it can be used.

Finally, this research has been able to demonstrate that clients can utilise MQTT bro-kers in a secure manner in order to transfer data to another MQTT broker, and it has also been able to assess the quality of the security in the data transmission. The location of MQTT brokers in respect to their clients must be taken into consideration in order for safe transmission to be established between them. All transmission notes in a transmission session that are involved in sending tasks that are related to a subject must be encrypted in order to produce a transmission session that can be monitored and confirmed. This is one of the most significant research accomplishments of this study.

Although these implications have been emphasised, the study has several limitations, particularly in terms of the extent of the network within the smarhome, which must be addressed. Typically, this is done to ensure that all of the tasks linked with a subject are effectively completed. A MQTT broker that has subscribed to the subject will require a secure transmission session in order to get updates about the topic, and will only receive updates about the topic that have been published in a secure transmission session. As a result, future studies should take into account the vast network environment.

- Interest to the Readers: The conclusions would surely interest the readers of the Sensors journal, and not only them, as secure IoT transmissions apply to various domains. The paper is interesting and will attract many researchers.  

Thank you very much for this. I am very grateful

- Overall Merit: The described solution can be implemented by other researchers and can be involved in other smart contracts experiments. 

Thank you very much for this. I am very grateful

- English Level: The level of English language is advanced. Through the entire paper, the language was appropriate and understandable, being easy to follow the flow since the beginning.

Thank you very much for this. I am very grateful

Reviewer 3 Report

Author in manuscript An Improved MQTT Secure Transmission Flags in a Smart Homes proposed a smart grid secured transmission flags suitable for preventing every bit of data transmission in a smart home. 

The presented research topic and problem are definitely relevant and welcomed in this type of journal. Nevertheless, there are several shortcomings that should be addressed:

  1. Research motivation, hypothesis, goal and purpose are not highlighted in the introduction.
  2. Figure 1 is very trivial and low quality
  3. Figure 2 is low quality and hard to interpret as well
  4. Figure 5 presents the user interface of a mobile app that has nothing related to the research. 
  5. Experimental analysis should have some experimental results that are not visible through this section. It is very theoretical and it is not related to scientific research. 
  6. Results are shown in the Patents section which is a very confusing title and it is not logical in a research paper. 

Paper has the potential but requires rewriting and restructuring with a focus on explaining research methodology and results interpretation. 

Author Response

Author in manuscript An Improved MQTT Secure Transmission Flags in a Smart Homes proposed a smart grid secured transmission flags suitable for preventing every bit of data transmission in a smart home. 

The presented research topic and problem are definitely relevant and welcomed in this type of journal. Nevertheless, there are several shortcomings that should be addressed:

  1. Research motivation, hypothesis, goal and purpose are not highlighted in the introduction

Research motivation has been addressed by

The fact that the MQTT protocol is one of the standardised protocols used in the Internet of Things for communication within an end-to-end network context in which security considerations would be crucial is the motivating factor for this research [4]. Furthermore, it was clear that many IoT developers used MQTT because of its features, such as its ability to run on a small amount of memory and less computing power, and its low bandwidth needs. Unfortunately, security and the integrity of data are still the main reasons why the protocol isn't used in many networks. People can also make new versions of this protocol because they can download and use the source code. This makes it easy to make new versions from the MQTT version 5 [5]. The MQTT protocol set up a subscriber-publisher interaction with the help of a broker in a transmission over TCP [6

Hypothesis has been addressed by

In light of this, the current research proposes the hypothesis that entails: "Transmission Flags" in a transmission session over the MQTT protocol are associated with responses to security events at varying degrees of sensitivity to detecting security issues

Goal and purpose has been addressed by

In general, the MQTT broker architecture operates as a middleman between a publisher and a subscriber in order to establish a link between the two. In order to maintain secure contact between subscribers and publishers, it is necessary to develop a good security technique that provides secure communication between both parties, which is the goal of this article.

  1. Figure 1 is very trivial and low quality

Figure 1 the MQTT publish-subscribe model has been improved

  1. Figure 2 is low quality and hard to interpret as well

Figure 2 The Conceptual Smart home MQTT model has been improved

  1. Figure 5 presents the user interface of a mobile app that has nothing related to the research. 

Figure 5 is the secure MQTT model indicating the secure package, while Figure 6 is an interface of the software application provided for transmission  

  1. Experimental analysis should have some experimental results that are not visible through this section. It is very theoretical and it is not related to scientific research. 

The experimental results are presented in section 6. Presentation of Experimental Analysis Results and Discussion

  1. Results are shown in the Patents section which is a very confusing title and it is not logical in a research paper. 

It is the section 6. Presentation of Experimental Analysis Results and Discussion

  1. Paper has the potential but requires rewriting and restructuring with a focus on explaining research methodology and results in interpretation. 

Interpretation of the results are provided in the last four paragraphs of section 6

Round 2

Reviewer 1 Report

> Actually I intend to present “Transmission Flags”

Well you spend 4 pages discussing an architecture for smart-homes... You should focus instead in what you propose.

The concept of "Transmission Flags" you claim to be the main subject of your paper is less than 1 page. It is also poorly described and very hard to read, I cannot understand from the text what novelty do you propose that doesn't already exist in MQTT implementations. Your "algorithm 1" is also poorly formatted and can't understand it ... But again, there is nothing novel in it you are just running MQTT over TLS, that's standard in all MQTT implementations I'm aware of.

That becomes even more clear in chapter 5, what changes have you made to Mosquitto broker ? and to the Eclipse PAHO library to support these "security flags" ?
It is of outmost importance that a reader can replicate your results, you don't provide information on the changes you made to existing open source software, not even a link to a public software repository.

Figure 6 is impossible to read! and Figure 7 should have the y-axis unit in the figure (not in the body of the text), still not clear what it represents... 4 different graphs, each with 10 sessions consisting of what? we need titles!

Table 1 is also still unclear and hard to read, how does it relate to figure 7 ?

Text received minor changes, it is still very hard to read and understand, might be the case some of my objections and comments are purely because the text is hard to read and understand.

Author Response

R1

Well you spend 4 pages discussing an architecture for smart-homes... You should focus instead in what you propose.

Thank you very much for sharing your thoughts, opinions, and perceptions on the relationship between the number of pages on the architecture and purpose of the work. This is something I take very seriously, and it is reasonable. The architecture is a component of my goal, and it describes the fundamental understanding of the "Computing Devices Network in Smarthome" from this point of views. All of the other sections of the paper, on the other hand, are a subset of the overall goal of this investigation. Therefore, within the architectures, the purposes/goal associated with research are put in which is why your next comment on the concept of "Transmission Flags".

The concept of "Transmission Flags" you claim to be the main subject of your paper is less than 1 page.

Thank you for taking the time to share your thoughts, opinions, and perceptions on the concept of "Transmission Flags" This concept is part of the networking system, and it entails where transmission will begin and where it will end. Throughout the papers, from the introduction to the conclusion, the flags serve as "indicators" of network activities that are described in greater detail within each section and that is why in your comment 1, you notice that the description of the architecture takes 4 pages, it is because transmission flags details are more described in the architecture 

It is also poorly described? and very hard to read, I cannot understand from the text what novelty do you propose that doesn't already exist in MQTT implementations.

Thank you for taking the time to share your thoughts, opinions, and perceptions on novelty. The "Transmission Flags within the transmission session of the sensors in smart home" is the primary concern of this paper. That is my novelty, I would be happy to read a paper that does similar work with similar approach, (because at this point, to my knowledge I am not aware of any paper that investigate "Transmission Flags within the transmission session of the sensors in smart home") and give inference on the dynamics of the security flags associated to those transmissions. To my little knowledge that is why I submit the paper to this reputable journal so that I can improve my potential.

Your "algorithm 1" is also poorly formatted and can't understand it ... But again, there is nothing novel in it you are just running MQTT over TLS, that's standard in all MQTT implementations I'm aware of.

Thank you for taking the time to share your thoughts, opinions, and perceptions on algorithm The representation of the algorithm is based on a line-by-line annotation, with the description of the algorithm appearing below the algorithm for each line of the algorithm that was represented. For this reason, the algorithm was written in such a way that the respectable readers of this journal could understand exactly what was going on in each line of the algorithm.

Thank you for taking the time to share your thoughts, opinions, and perceptions on MQTT over TLS In response to your comment: “Running MQTT over TLS, that's standard in all MQTT implementations I'm aware of” running MQTT over TLS and SSL, is one of the standard in a secure MQTT implementations however the security situation can be disputed, that is why, those transmission flag responsible for signalling the security issues are a concern to this study.

That becomes even more clear in chapter 5, what changes have you made to Mosquitto broker? and to the Eclipse PAHO library to support these "security flags”?

Thank you for taking the time to share your thoughts, opinions, and perceptions on Mosquitto broker, and Eclipse PAHO. I think you have a vast knowledge on this, for me I don’t think if I could remember indicating that I have change from the sources (open source). in terms of implementing Mosquitto broker, what I presented is: “When it came to communication between devices and apps, the MQTT Protocol was used to operate over the transport standard for communication. Open SSL, which is a protocol for delivering encrypted connection between devices and applications, was also used to run the design. The implementations were carried out using Eclipse Paho, whereas Eclipse Mosquitto is a lightweight server implementation of the MQTT protocol that is suited for a wide range of circumstances, from full-power workstations to embedded and low-power devicesfurthermore, I will like to humble put it clearer, that I did not make or create the security flag, I run several transmissions and I reported 10 of them here, which are all the transmission sessions over MQTT(secure) and I capture those session data that I descriptively analysed and draw conclusion.  It is not that I alter with the transmission, I set it up and runs it and captured data were I analysed and draw conclusion.

It is of outmost importance that a reader can replicate your results, you don't provide information on the changes you made to existing open source software, not even a link to a public software repository.

Thank you for taking the time to share your thoughts, opinions, and perceptions on open source. I think to highly advanced researchers like you, you will much appreciated “adding or subtracting” from the open source. In this current work I setup a link-to-link over secure MQTT and capture the transmission sessions within their scenarios. It’s not the development that is my novelty, but the experimental analysis. I didn’t change the open source software. 

Figure 6 is impossible to read! and Figure 7 should have the y-axis unit in the figure (not in the body of the text), still not clear what it represents... 4 different graphs, each with 10 sessions consisting of what? we need titles!

Thank you for taking the time to share your thoughts, opinions, and perceptions on the figures. Figure 6 is now occupying a single page and clear to read. Figure 7 now have the y-axis unit in the figure, which represent the experimental scenarios for the tranmissions on various transmission session, now they are labelled A, B, C, D and the description of each according to the title A, B,C,D was given

Table 1 is also still unclear and hard to read; how does it relate to figure 7?

Thank you for taking the time to share your thoughts, opinions, and perceptions on the table. The size of Table 1 is increased and now is clear to read, it relates to figure 7 because it provides the details data within each session(s) that was presented in figure 7

Reviewer 2 Report

I thank the author for modifying the paper according to my comments. Unfortunately, the images remained of low quality. Can you please add higher resolution images? Thank you!

Author Response

R2

I thank the author for modifying the paper according to my comments. Unfortunately, the images remained of low quality. Can you please add higher resolution images? Thank you!

Thank you for taking the time to share your thoughts, opinions, and perceptions on images. I added higher resolution images

Reviewer 3 Report

Suggestions and comments are addressed well, with exception of improving image quality (they remained low) and presenting more clearly research methodology (trough phases, activities, etc.)

Author Response

R3

Suggestions and comments are addressed well, with exception of improving image quality (they remained low) and presenting more clearly research methodology (trough phases, activities, etc.)

Thank you very much for the comment, I make section 3 to be the research methodology and described the phases